# A Data-Driven Digital Application to Enhance the Capacity Planning of the COVID-19 Vaccination Process

**DOI:** 10.3390/vaccines9101181

**Published:** 2021-10-15

**Authors:** Berend Markhorst, Tara Zver, Nina Malbasic, Renze Dijkstra, Daan Otto, Rob van der Mei, Dennis Moeke

**Affiliations:** 1Department of Mathematics, Vrije Universiteit Amsterdam, 1081 HV Amsterdam, The Netherlands; b.t.markhorst@student.vu.nl (B.M.); a.t.zver@student.vu.nl (T.Z.); n.malbasic@student.vu.nl (N.M.); r.k.dijkstra@student.vu.nl (R.D.); d.otto@student.vu.nl (D.O.); Rob.van.der.Mei@cwi.nl (R.v.d.M.); 2Stochastics Group, Center for Mathematics and Computer Science, 1098 XG Amsterdam, The Netherlands; 3Research Group Logistics & Alliances, HAN University of Applied Sciences, 6826 CC Arnhem, The Netherlands

**Keywords:** COVID-19, vaccination planning, vaccination logistics, last-mile, decision support system

## Abstract

In this paper, a decision support system (DSS) is presented that focuses on the capacity planning of the COVID-19 vaccination process in the Netherlands. With the Dutch national vaccination priority list as the starting point, the DSS aims to minimize the per-class waiting-time with respect to (1) the locations of the medical hubs (i.e., the vaccination locations) and (2) the distribution of the available vaccines and healthcare professionals (over time). As the user is given the freedom to experiment with different starting positions and strategies, the DSS is ideally suited for providing support in the dynamic environment of the COVID-19 vaccination process. In addition to the DSS, a mathematical model to support the assignment of inhabitants to medical hubs is presented. This model has been satisfactorily implemented in practice in close collaboration with the Dutch Municipal and Regional Health Service (GGD GHOR Nederland).

## 1. Introduction

The coronavirus pandemic has been holding the world in its grip for more than a year now. Since the start of the outbreak, the virus has not only posed immense challenges to healthcare systems worldwide but also to society. Although some serious measures have been taken to minimize exposure and transmission of the coronavirus, widespread vaccination is believed to be a safe and critical step in ending the pandemic. Different parties are working hard to produce the necessary vaccines, most of which are already sufficiently available in the EU countries.

However, how do we get the available vaccines fast, carefully and responsibly from ‘lab to arm’? The delivery of COVID-vaccines is a complex logistic challenge. According to [1], the last-mile delivery is one of the most critical parts of the COVID-19 vaccine supply chain. It is this last-mile challenge that is addressed in this paper. More specifically, a data-driven digital application to support the capacity planning of the last-mile vaccination process in the Netherlands is presented and discussed.

When it comes to the streamlining of the last-mile COVID-19 vaccination process in the Netherlands, it is required to distinguish between “vulnerable persons” and “less vulnerable persons”. This is because, from a logistics perspective, the needs and requirements differ vastly. Vulnerable persons, which have higher priority, are vaccinated in their own homes, hospitals, or nursing homes where they stay. Less vulnerable persons are administered via large-scale vaccination centers, which have been set up in places like sport centers or conference halls. In this study, the focus lies on the vaccination of less vulnerable persons via large-scale vaccination centers, which from now on will be referred to as (medical) hubs. Furthermore, in line with the advice of [2,3] and the desired approach of the Dutch government, it is assumed that vaccination sites are part of an overall plan, rather than operating independently. Hence, from a capacity planning perspective, this overall approach is important in order to (1) avoid sub-optimal allocation of resources, and (2) to enhance an equitably and justly distribution of the vaccines, as emphasized to be of importance by e.g., [4]. Finally, given the significant social impact of the COVID-19 vaccination program, the processes and underlying progress should be as transparent as possible (see, for example, [3,5]).

Since the start of the COVID-19 outbreak, only a few studies have appeared in the literature which investigate the (logistics) planning of the last-mile of the COVID-19 vaccination process. In [6], a hybrid simulation model of a drive-through mass vaccination clinic is introduced. The model can be used to enhance the planning, design, operation, and feasibility and effectiveness assessment of such facilities. The authors of [7] present a mathematical programming model to enhance more equitable distribution of COVID-19 vaccines in developing countries, whereby taking into account the needs for cold, very cold, and ultra-cold supply chains. The study of [8] focuses on the planning and location of COVID-19 vaccination sites. They propose a mathematical programming model for selecting the optimal vaccination sites out of a given set (i.e., university hospitals, health department related locations, and general practices). Different patient-to-facility assignments and doctor-to-facility assignments and different constraints on the number of vaccines per site or maximum travel time are used. In [9] the distribution of vaccines to vaccination sites is optimized using Geographic Information Systems (GIS) technology. By applying the developed methodology in the city of Warsaw in Poland, the authors demonstrated ways to achieve uniform vaccination coverage throughout the city. In the recent study of [10], a simulation-based approach is presented which combines route optimization and dynamic simulation to improve the logistics performance of the COVID-19 vaccine distribution. The result of their scenario analysis shows that service level, cost-effectiveness, environmental performance, and equity of a cold chain vaccine logistics system can be significantly influenced by the fleet size, the fleet composition, the type of vehicle used, and the route optimization. Another study [11] developed a novel control framework to derive optimal vaccination strategies, whereby considering epidemiological projections and constraints on vaccine supply and distribution logistics. Although, in the past, numerous studies have been published regarding the planning of the last-mile of large scale vaccination programs (see, e.g., [12,13,14,15]), these approaches are not readily applicable for the streamlining of the COVID-19 vaccination process [16].

### 1.1. Contribution

Based on our literature findings and the recent overview paper of [17], we conclude that studies that focus on streamlining the last mile of the COVID-19 vaccination programs are scarce. To the best of our knowledge, this paper is among the first to present a model and a DSS, which addresses the challenge of: minimizing the per-class waiting time (based on the national “vaccination priority list”) with respect to (1) the locations of the medical hubs (i.e., the vaccination locations) and (2) the distribution of the available vaccines and healthcare professionals between the medical hubs (over time). The DSS presented in this paper is ideally suited for providing support in a continuously changing environment. Besides providing support for the short-term (dynamic) planning, the DSS can assist policy makers in getting a better understanding of the possible consequences of (restricting) the availability of key-resources (e.g., vaccines, vaccination capacity, and number of medical hubs).

Although the Dutch situation is taken as a point of departure in this paper, the proposed approaches and DSS could be applied to other countries or regions in a similar fashion.

### 1.2. Problem Description

In the Netherlands, the National Institute for Public Health and the Environment (RIVM) (https://www.rivm.nl, accessed on 1 May 2021) has published a priority list that assigns different groups of Dutch inhabitants to different vaccines and vaccination moments. For example, persons between 18 and 60 years old without a medical condition have the lowest priority. Hence, different priority classes can be defined for receiving their vaccinations at the medical hubs. An important performance measure in this context is the per-class waiting time distribution. In other words:


*x% of persons in priority class A have to wait y time units to be vaccinated.*


Note that these distributions may also be location-dependent. For example, the waiting times in urban areas may differ from rural areas. In addition, the performance of the last mile of the vaccination program is also influenced by:The number and locations of the medical hubs.The available vaccination capacity of a medical hub.The number of healthcare professionals (authorized to administer the vaccine).The number of available vaccines per medical hub.

In this paper, we present a model that optimizes the aforementioned performance measure of per-class waiting time. This is done by determining the optimal placement of medical hubs as well as the division of vaccines and healthcare personnel. Secondly, with the optimization model as core input, a user-friendly DSS is presented which provides the user with the possibility of conducting “what-if” analysis. As part of this what-if functionality, the DSS requires the user to define several input parameters of which an overview can be found in Section 3. In addition to the DSS, in Section 4, a mathematical model to support the assignment of inhabitants to medical hubs is presented. This model has been successfully implemented in practice in close collaboration with the Dutch Municipal and Regional Health Service (GGD GHOR Nederland). This paper closes with some conclusions and a short discussion.

## 2. Modelling Approach

The underlying model of the DSS can roughly be divided into the following processes:Determining the optimal locations of the medical hubs.Determining the optimal distribution of the medical capacity (i.e., the available vaccines and healthcare workers) over the medical hubs.Creating a simulation of the vaccination process.

Concerning the COVID-19 vaccination process, it is difficult to answer questions strictly mathematically. For example, different persons might answer differently to questions such as: “To what extent should healthcare workers get more prioritization?”. Therefore, the user is given the freedom to experiment with different starting positions and strategies (i.e., conduct “what-if” analysis). In addition, with respect to the development of the algorithms, trade-offs had to be made between computation time and accuracy. Hence, the DSS should be sufficiently accurate and at the same time provide the user with a smooth interactive experience.

### 2.1. Location of the Medical Hubs

When it comes to the placement of the medical hubs, the facility location problem [18], set covering problem [19], and COVER-CAP [20] are well-known formulations (and extensions) of the problem at hand. As these approaches are all based on Integer Linear Programming (ILP), the state space suffers from the curse of dimensionality. Consequently, the computation time of these approaches is prohibitively long given a large number of constraints and decision variables as they occur in practice. As the problem at hand could not be optimally solved within an acceptable time frame, a fast yet accurate heuristic was required. The developed heuristic resembles the divide-and-conquer structure discussed by [21]. Based on this structure, we partitioned the problem-solving process into two steps. In the first step, we assume that each municipality has its own hub. This step is referred to by [21] as the dividing step. In the second step, we try to merge hub locations. More specifically, hub locations are merged if the distance constraints can be met; otherwise the municipality will get its own hub. The merging of hub locations corresponds with the so-called conquer-step of [21]. When executing this algorithm, it is ensured that each municipality is covered. See Algorithm 1 for the pseudo-code of the hub-placement algorithm.

Two limitations should be mentioned with respect to the hub-placement algorithm:As we use heuristic Euclidean distances between the centroids of municipalities, the “true” travel distance, which can be calculated by using route planners (depending on the travel modality), may exceed the maximum travel distance.The heuristic does not take the costs concerned with opening a hub into account. This was a design decision, as these costs were unknown beforehand. However, as the algorithm aims to minimize the number of hubs, it does work in favor of lowering the hub opening costs.

The verification of the hub placement is performed via the placement of circles, with a radius of the maximum travel distance, on the chosen hubs. If the areas of all circles cover the centroids of all the municipalities, we can conclude that the algorithm has found a feasible solution. An example is provided in Figure 1.
**Algorithm 1.** Compute locations of the hubs.1: every municipality has its own hub.2: **while** not every hub is done **do**3:  select the smallest hub that is not done yet (e.g., hub A)4:  find the nearest hub to the smallest hub (e.g., hub B)5:  **if** the maximal distance constraint is violated **then**6:   hub A is done7:  **else**8:   **if** distance constraint between center hub A and all municipalities in hub B are not violated **then**9:    merge municipalities hub B in hub A10:    remove hub B from hub list11:   **else if** distance constraint between center hub B and all municipalities in hub A are not violated **then**12:    merge municipalities hub A in hub B13:    remove hub A from hub list14:   **else**15:    hub A is done16:   **end if**17:  **end if**18: **end while**

### 2.2. Distribution of Medical Capacity

Regarding the distribution of medical capacity, it is assumed that the order in which priority groups should be vaccinated is given and can also be altered by the end-user. For the allocation of medical capacity, two different approaches were implemented in the DSS:*Equal distribution of vaccines across the hubs*—The available number of vaccines per day is equally distributed across the hubs that are not finished with vaccinating yet. Thus, if on a given day, *H* hubs are not finished with vaccinating, and there are *V* vaccines available for that day, then each of those *H* hubs will get the same number of vaccines, namely:
$$\left\lfloor \frac{V}{H}\right\rfloor$$The advantage of this method is that each hub will receive the same number of vaccines, independent of its size. As a result, smaller hubs will be finished vaccinating earlier than larger hubs. Consequently, larger hubs will eventually get more vaccines per day once the number of finished hubs increases and are no longer considered in the division of vaccines. A major disadvantage of this method is that, during the early stage, densely populated areas will not receive more vaccines, which increases the risk of a local COVID-outbreak.*Proportional allocation of vaccines over hubs*—This strategy is based on the number of susceptible persons. For this strategy, the number of non-vaccinated persons (i.e., susceptible persons) in the whole country or region, $S_{\mathrm{total}}$, and the number of non-vaccinated persons for each hub, $S_{h}$, are determined. The number of available vaccines per day is denoted as *V*. To calculate the number of vaccines that each hub gets per day, the following formula is used:
$$\left\lfloor \frac{S_{h}}{S_{\mathrm{total}}*V}\right\rfloor$$Over time, every hub will get the same number of vaccines, as the proportional distribution has not been altered. As a result, every hub will be finished vaccinating almost simultaneously. An advantage of using this proportional allocation is that the areas that are more likely to endure local outbreaks (due to a high number of non-vaccinated persons) will receive more vaccines.As both approaches have been implemented for the allocation of medical capacity, which includes vaccines and health professionals, a short summary can be found below that discusses each separately.*Vaccine allocation*—The pseudo-code of the vaccine allocation algorithm is shown in Algorithm 2. The implementation consists of two parts: *the allocation* and *the number of shots required*. Being able to deal with both one- and two-shot vaccinations requires more advanced algorithms. For example, once a person is vaccinated for the first time, another shot from today’s supply should be reserved for the person to be given later to make sure it will be available when needed. To support this additional feature, the time period between the first and second shot has been added as an additional input parameter. In this way, it is ensured that there are enough vaccines available to provide persons with their second shot, independently of the timing when this would take place. The reason for choosing this approach is to make the model more realistic as it resembles the strategy of the Dutch government.*Healthcare worker allocation*—Algorithm 3 shows the pseudo-code of the healthcare worker allocation algorithm. Before applying the equal and proportional approach, which is similar to the vaccine allocation, a distinction should be made between a single- or double-shot vaccine. In case of the double shot vaccine, the algorithm reserves sufficient capacity for the second shot. Once the required “second shot capacity” has been determined, the remaining number of healthcare workers get divided over the hubs either proportionally or equally.

The distribution of vaccines and healthcare workers and its effect on the vaccination timeline has been verified numerically (comparison between numerical computations and the algorithm’s outcomes).
**Algorithm 2.** Division of vaccines.1: With some divisions, it is necessary to floor (**) the resulting numbers.2: **for** every hub **do**3:  **if** allocation is proportional **then**4:   Total left over: compute the national number of people that are left over from all priority classes5:   Hub left over: compute the number of people that are left over from all priority classes on a hub-level6:   **if** Hub left over is 0 **then**7:    Hub is done vaccinating new people.8:   **else**9:    (**) Amount of vaccines: hub left over / total left over * number of vaccines available10:   **end if**11:  **else if** allocation is equal **then**12:   Left over hubs: compute the number of hubs that are still busy vaccinating new people13:   **if** hub is still busy vaccinating new people **then**14:    (**) Amount of vaccines: number of vaccines available / left over hubs15:   **else**16:    Hub is done vaccinating new people.17:   **end if**18:  **end if**19: **end for**

**Algorithm 3.** Division of healthcare workers.1: With some divisions, it is necessary to floor (**) the resulting numbers and with others to ceil (*).2: **if** two shots are necessary **then**3:  **for** every hub **do**4:   Second shot: check how many people come for a second shot.5:   (*) Minimal amount of healthcare workers: second shot / (8 hours * vaccination speed per hour per healthcare worker)6:  **end for**7: **end if**8: Compute the remaining number of available healthcare workers.9: **for** every hub **do**10:  **if** allocation is proportional **then**11:   Total left over: compute the national number of people that are left over from all priority classes12:   Hub left over: compute the number of people that are left over from all priority classes on a hub-level13:   (**) Amount of healthcare workers: hub left over / total left over * number of healthcare workers available14:  **else if** allocation is equal **then**15:   Left over hubs: compute the number of hubs that are still busy vaccinating new people16:   **if** hub is still busy vaccinating new people **then**17:    (**) Amount of healthcare workers: number of healthcare workers available / left over hubs18:   **else**19:    Amount of healthcare workers: 020:   **end if**21:  **end if**22: **end for**

### 2.3. The Vaccination Process

In the previous sections, the approach for determining the hub locations and allocation of healthcare workers and vaccines across these hubs has been described. The final step in the model involves the *vaccination process*. More specifically, this process is a simulation of a single ’vaccination-epoch’ that enlists the vaccination of the *willing* members of all priority classes. Note that the persons who are not willing to get vaccinated will not be considered. To compute the number of persons that can get a vaccine, the total vaccination time is computed as follows:

$$T_{\mathrm{total}}=\min\left\{N\cdot R,V\right\},$$
where $T_{\mathrm{total}}$ is the number of people that can get a vaccine, *N* is the number of available healthcare workers, *R* is the average number of vaccines a healthcare worker can administer in a day and *V* is the number of vaccines available that day. The reason for taking this approach is that if there are less vaccines than healthcare workers available, the number of vaccines becomes the bottleneck. If not, then the “healthcare worker capacity” becomes the bottleneck. Note that if a person requires two doses of the vaccines, *the number of available vaccines* is divided by two to ensure availability for the second shot. This process is repeated until the first priority group is fully vaccinated. After that, the vaccination process of the second priority group starts. In the meantime, the model keeps track of the number of persons getting their first or second shot, when they received the shot and how many vaccines and healthcare workers were needed per hub. This process is repeated until all applicable persons, willing to get a vaccine, have been vaccinated.

## 3. Decision Support System

In this section, we present a DSS that can be used to evaluate alternative strategies by allowing the user (in a user-friendly manner) to change its input parameters. User-friendliness was a key element during the development of the dashboard of the DSS. Hence, in the development of the dashboard, particular attention has been paid to the clarity, intuitiveness, and response time. As mentioned in Section 2, the key in this context is the per-class waiting time distribution, defined as *x% of persons in priority class A have to wait y time units to be vaccinated*. See https://lab-to-arm.com (accessed on 1 September 2021) for an online version of the DSS.

### 3.1. Input Parameters

To allow the user to interactively explore a wide variety of what-if scenarios, several input parameters must be tuned to match the required what-if situation. The DSS requires the following input parameters to be specified by the user:*The number of available vaccines per period:* To include uncertainty regarding the number of available vaccines, one can dynamically fill in the number of available vaccines per day per period. Thus, different amounts can be used for various dates.*The number of vaccines per hour per healthcare worker:* The (average) number of vaccines that can be administered by one healthcare worker per hour, given that a working day is 8 h long.*The number of healthcare workers per day:* The total number of available healthcare workers per day. These healthcare workers will be distributed across the hubs according to the selected allocation strategy.*The allocation of healthcare workers and vaccines:* Two different allocation strategies can be chosen from: ‘equal’ and ‘proportional.’ By choosing the option ’equal’ the available number of healthcare workers and vaccines will be equally distributed across all hubs. On the other hand, by choosing ’proportional’ the available number of healthcare workers and vaccines will be divided between the hubs according to the proportion of inhabitants that still need to be vaccinated in that hub.*The vaccination strategy prioritization:* The Dutch government has defined priority classes. For this model specifically, only three have been considered: healthcare workers, non-vulnerable elderly (65+), and non-vulnerable adults (18–65). However, this input parameter has been implemented to account for the continuously changing order of vaccination prioritization. It enables the user to switch the order in which these three priority classes should be vaccinated.*The number of vaccine shots:* The user can select whether one or two shots (doses) are required. As the majority of available vaccines in the Netherlands require two shots, the latter option is set as the default value.*The number of days between the shots:* This is an essential input parameter, as the number of days between two shots can vary between vaccines. In addition, it should be mentioned that once a person receives the first shot (of a two-shot vaccine), the second shot is instantly saved and is not given to anyone else. This ensures that each person who is vaccinated once gets the other shot, regardless of the storage of available vaccines.*The maximum acceptable travel distance:* The maximum distance (in kilometers) that persons are willing to travel to receive their vaccine.*The coverage ratio:* The proportion of the Dutch population that should be vaccinated.*The willingness to get vaccinated:* The proportion of persons that is willing to get vaccinated (per priority class).

It should be noted that each input parameter has a predefined default value.

### 3.2. Output

The dashboard of the DSS provides the user with the following information:The total number and location of the (medical) hubs.The overall waiting time.The waiting times per priority class (healthcare workers, elderly, and adults).

Moreover, the user can gather insights from different graphs shown by using the tabs *Overview* and *More detailed information*.

#### 3.2.1. Overview Tab

The first three visuals discussed in this tab cover the guidelines for policymakers to decide where and how many medical hubs should be placed. These visuals are closely related to the maximum travel distance parameter, which the user can adjust in the input section.

*Location of the hubs:* This graph shows the placement of the medical hubs in the Netherlands, where each point denotes a single hub. The user can hover over the hubs, which will provide each medical hub’s location.*Area of the hubs:* The user is provided with the option to see which municipality belongs to which medical hub, as multiple municipalities can belong to a single hub (see Figure 2). In addition, by hovering over the map the user can see more detailed information about each municipality.*Impact travel distance:* The user can see the effect that the maximum travel distance has on the number and location of the medical hubs.

Furthermore, the bottom two visuals in the *Overview*-tab provide more insight into the actual vaccination process.

*Vaccination process over time (graph):* The graph shown in Figure 3 provides detailed information on the fraction of vaccinated persons over time for each priority class. The exact information on when a particular priority class is finished with vaccinating is also given by the KPIs at the top of the page.*Vaccination process over time (animation):* Closely related to the graph mentioned above, an animation of the vaccination process over time can be found.

#### 3.2.2. More Detailed Information Tab

The dashboard’s final tab provides the user with insight on how the chosen starting position and vaccination strategy affect the allocation of vaccines and healthcare workers across the hubs (over time).

*Division of healthcare workers/vaccines:* By clicking the *Select division* button, the user can choose between healthcare workers and vaccines. The visual (see Figure 4) shows how the vaccines are allocated across the medical hubs over time.*The fraction of vaccinated persons:*Figure 4 shows the division of vaccines across hub over time on an aggregate (i.e., national) level. However, the dashboard also provides the user with in-depth information on hub level.*The fraction of persons with the first shot (active):* This graph, shown in Figure 5, will only appear when the option “two shots” is selected. It shows, for each priority class, the fraction of persons who have received their first vaccine shot over time on a national level.

## 4. Implementation in Practice

Besides the aforementioned topics, a project was started that aimed to use mathematical approaches to distribute vaccines in a more efficient manner over the medical hubs. This was performed in close collaboration with the Dutch Municipal and Regional Health Service (GGD GHOR Nederland) https://ggdghor.nl, accessed on 10 September 2021, which is responsible for the organization and execution of large-scale vaccination programs. A mathematical model that assigns inhabitants to medical hubs was developed, which is described in more detail in Section 4.1. The model has been tested in practice and shown to be useful for the further improvement of the last-mile of the COVID-19 vaccination process.

### 4.1. Assigning Inhabitants to Medical Hubs

In this section, an approach to assigning inhabitants to the medical hubs is further described. It is assumed that inhabitants want to travel to the closest hub. To meet this assumption, we propose an approach, in which inhabitants are assigned based on their postal code. For the problem under hand, an ILP has been constructed, consisting of the following sets, parameters and decision variables. *P* is the set of postal codes. *H* is the set of hubs. *D* is the maximum distance. $d_{ij}$ is the distance between postal code i∈P and hub j∈H. $a_{ij}$ as $I_{\left\{d_{ij}\leq r\right\}}\;\forall i\in P,j\in H$ denotes whether hub *j* can be reached from postal code *i* by not exceeding the maximum travel distance *r*. As for the decision variables, $x_{ij}$ is used, which denotes whether postal codes i∈P is assigned to hub j∈H. In a more concrete form, the ILP as displayed in (Equation 1) is constructed.
(1)$$\begin{array}{cccc} \mathrm{minimize} & \sum_{i\in P}\sum_{j\in H}d_{ij}x_{ij} & \\ \mathrm{subject}\;\mathrm{to} & x_{ij}\leq a_{ij} & \;\forall i\in P,j\in H, & \\ \; & x_{ij}\in \left\{0,1\right\} & \;\forall i\in P,j\in H, & \\ \; & \sum_{h\in H}x_{ij}=1 & \;\forall i\in P \end{array}$$

It should be noted that this part of the overall model has not (yet) been implemented in the online version of the DSS.

## 5. Conclusions Discussion

We presented a Decision Support System (DSS) that supports decision making regarding the capacity planning of the COVID-19 vaccination process. The DDS aims to minimize the per-class waiting time with respect to the vaccination locations and the distribution of the vaccines and healthcare professionals across large-scale vaccination locations. Furthermore, the DSS-user is free to experiment with different starting positions and strategies, and as such, the DSS is ideally suited for providing support in a dynamic environment. Besides providing support for the short-term (dynamic) planning, the DSS can assist policymakers in getting a better understanding of the possible consequences of (restricting) the availability of key-resources (e.g., vaccines, vaccination capacity, and number of medical hubs). The successful implementation in practice of the model presented in Section 4.1 shows that mathematical models can be useful in streamlining the last-mile of the COVID-19 vaccination process.

We emphasize that the approaches and DSS presented in this paper are also applicable way beyond the Dutch context. Hence, they can be applied to other countries, regions and/or large-scale vaccination campaigns worldwide. This is because the model is highly generic and the main characteristics of the last-mile (logistics) infrastructure are often similar. Nevertheless, the current version of the DSS is particularly useful in a dynamic context where circumstances are continuously changing (e.g., availability of vaccines of different types, changes in supply, the availability of healthcare professionals). Despite being a valuable tool, some important limitations should be acknowledged. Transportation and storage constraints are not considered. In addition, the applied approach assumes a smooth vaccination process. Hence, stochasticity regarding, e.g., last-mile delivery times and the vaccination procedure itself are not explicitly taken into account.

Finally, we address a number of topics for follow-up research. Since the DSS in its current state uses relatively simple but fast heuristics, there is ample room for refinement of the model by applying meta-heuristics. Meta-heuristics are general-purpose optimization algorithms [22], where the term *meta* refers to the higher-level general methodologies, which are used to guide the underlying heuristic strategy [23]. Meta-heuristics are highly diverse in nature, see [24,25,26] and references therein for overviews. In the context of the present paper, a highly promising type of meta-heuristics is offered by the concept of greedy randomized adaptive search procedures (GRASP), which is based on an iterative approach consisting of two phases: (1) a *construction* phase, and (2) a *local search* phase. We refer to [27] for details on GRASP.

In addition to applying meta-heuristics techniques, it would also be interesting to incorporate uncertainty with respect to, e.g., the availability and amount of specific vaccine types, vaccination willingness, and the available healthcare workers into the model. This will raise the need for the development of forecasting models and stochastic optimization models and solution techniques.

## Figures and Tables

**Figure 1 vaccines-09-01181-f001:**
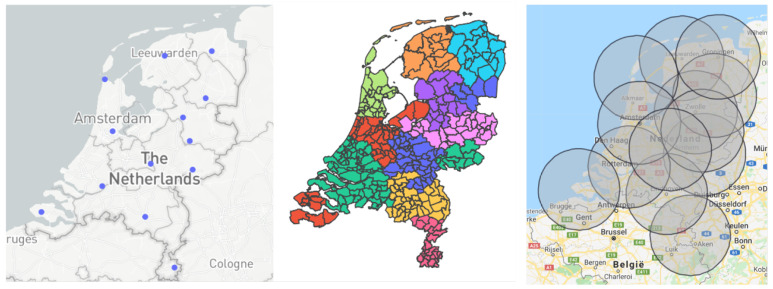
Verification of the placement of hubs by checking the maximum travel distance constraint (in this example 60 km).

**Figure 2 vaccines-09-01181-f002:**
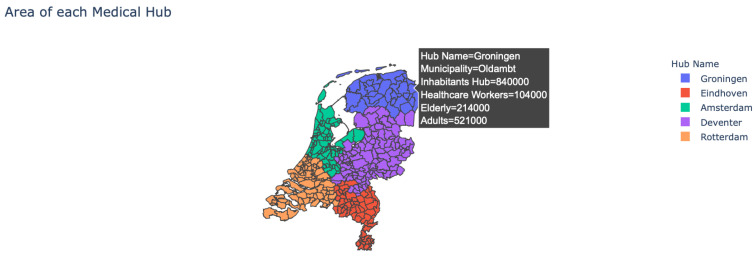
Overview of the location and number of medical hubs when applying the model to the Netherlands.

**Figure 3 vaccines-09-01181-f003:**
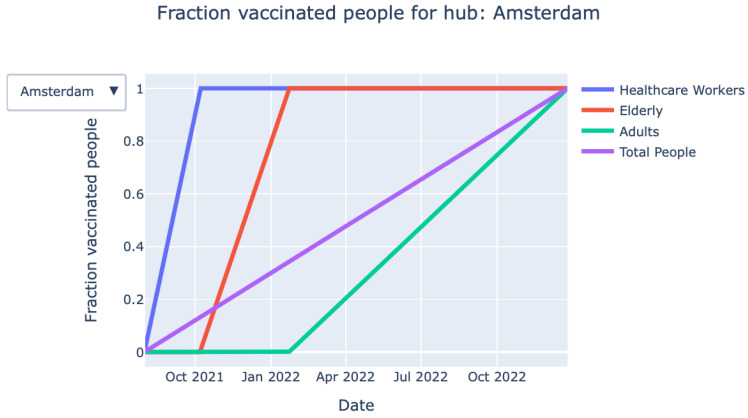
Fraction of vaccinated persons over time for each priority class.

**Figure 4 vaccines-09-01181-f004:**
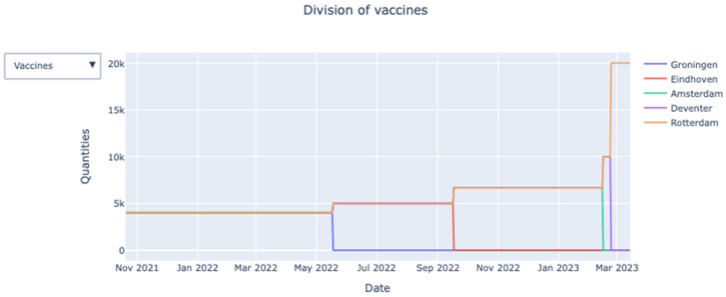
Allocation of vaccines across the medical hubs over time.

**Figure 5 vaccines-09-01181-f005:**
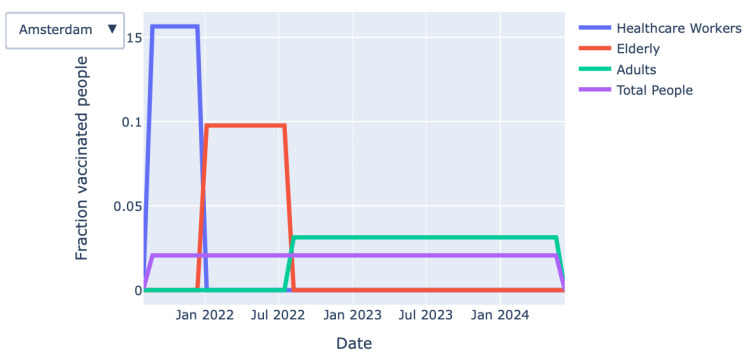
The fraction of persons who have received their first vaccine shot over time on a national level (for each priority class).

## Data Availability

The data of this study are available on request from the correspond ing authors.

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
