# Peer review of "A Data-Driven Digital Application to Enhance the Capacity Planning of the COVID-19 Vaccination Process"

_vaccines, 2021, doi:10.3390/vaccines9101181_

Round 1

Reviewer 1 Report

In the current period, the article is highly relevant. The methodology is adequate and replicable, with few published works. The text needs a language revision. The graph in Figure 4 shows continuous behaviour over time and is apparently unnecessary. The article comes to a sudden end. The results presented in topic four can be improved, there is a lack of discussion, and the conclusion presents the same sentences as the introduction. I can think of some discussion questions. What are the advantages and disadvantages of the model? How do the authors imagine the application of the model by other countries? How might the model be appropriate for other diseases? The authors could discuss the existing literature on large-scale vaccination programs (also considering other similar studies that are for other diseases). Suggested articles for discussion:

Rocha, T. A. H., Boitrago, G. M., Mônica, R. B., Almeida, D. G. D., Silva, N. C. D., Silva, D. M., ... & Vissoci, J. R. N. (2021). National COVID-19 vaccination plan: using artificial spatial intelligence to overcome challenges in Brazil. Ciencia & saude coletiva, 26, 1885-1898.

Krzysztofowicz, S., & OsiÅ„ska-Skotak, K. (2021). The Use of GIS Technology to Optimize COVID-19 Vaccine Distribution: A Case Study of the City of Warsaw, Poland. International Journal of Environmental Research and Public Health, 18(11), 5636.

Sun, X., Andoh, E. A., & Yu, H. (2021). A Simulation-Based Analysis for Effective Distribution of COVID-19 Vaccines: A Case Study in Norway. Transportation Research Interdisciplinary Perspectives, 100453.

Lemaitre, J. C., Pasetto, D., Zanon, M., Bertuzzo, E., Mari, L., Miccoli, S., ... & Rinaldo, A. (2021). Optimizing the spatio-temporal allocation of COVID-19 vaccines: Italy as a case study. medRxiv.

Minor corrections

Line 16 - “And although some” I suggest “Although some”

Line 21 - “But how“ I suggest “However, how”

Line 28 - “required to make a distinction” I suggest “required to distinguish”

Line 31 - “own homes, in hospitals or the nursing homes” I suggest “own homes, hospitals, or nursing homes”

Line 23 - “In this study the focus lies” I suggest inserting a comma “In this study, the focus lies”

Lines 40-42 - “Finally, given the large social impact that the COVID-19 vaccination program has, it is important that that the underlying processes and progress are as transparent as possible (see e.g., [3,5]).” I suggest “Finally, given the significant social impact of the COVID-19 vaccination program, the processes and underlying progress should be as transparent as possible (see, for example, [3,5]).”

Line 46 - “In [6] a hybrid” I suggest inserting a comma “In [6] a hybrid”

Line 48 - “Tavana et al. [7]” I suggest standardizing the quote. Sometimes the quote does not have the authors as in the previous sentence.

Line 49 - “enhance a more equitable distribution” I suggest “enhance more equitable distribution”

Line 56 - “And although, in the past numerous” I suggest inserting a comma “Although, in the past, numerous”

Line 62 - “which focus on the streamlining of the last mile of the COVID-19 vaccination programs are scarce.” I suggest I suggest “that focus on streamlining the last mile of the COVID-19 vaccination programs are scarce.”

Line 70 - “policy makers” I suggest “policymakers” (check the entire text)

Lines 70-71 - “understanding into the possible” I suggest “understanding of the possible”

Line 73 - “And although in this paper the Dutch situation is taken as point of departure” I suggest “Although the Dutch situation is taken as a point of departure in this paper,“

Line 76 - “list which assigns different” I suggest “list that assigns different”

Line 86 - “in the urban areas” I suggest “in urban areas”

Line 109 - “With regard to the COVID-19 vaccination process it is” I suggest “Concerning the COVID-19 vaccination process, it is”

Line 119 - “well known” I suggest “well-known”

Line 121 - “Programming (ILP) the” I suggest inserting a comma “Programming (ILP), the”

Line 122 - “the computation time of these approaches are” “the computation time of these approaches is”

Lines 122-123 - “the large number” I suggest “a large number”

Lines 140-141 - “However as the algorithm” I suggest inserting a comma “However, as the “

Line 145 - “we can conclude the algorithm” I suggest “we can conclude that the algorithm”

Line 152 - “be finished with vaccinating earlier” I suggest “be finished vaccinating earlier”

Line 171 - “professionals a short” I suggest inserting a comma “professionals, a short”

Line 187 - “In case of a” I suggest “In case of the”

Line 209 - “by two, in order to ensure” “by two to ensure”

Lines 154-155 - “and are not considered in the division of vaccines anymore.” I suggest “and are no longer considered in the division of vaccines.”

Line 2016 - “In this section we present a DSS which can” I suggest “In this section, we present a DSS that can”

Line 2019 - “of the dashboard particular attention” I suggest inserting a comma “of the dashboard, particular attention”

Lines 245-246 - “However, to account for the continuously changing order of vaccination prioritization, this input parameter has been implemented.” However, this input parameter has been implemented to account for the continuously changing order of vaccination prioritization.”

Line 251 - “as default value” I suggest “as the default value”

Line 295 - “strategy effects” I suggest “strategy affect”

Line 320 - “to assign” I suggest “to assigning”

Line 324 - “In this paper an” “In this paper, an”

Line 324 - “presented which supports” “presented, which supports”

Line 327 - “time with respect to the” “time to the”

Same phrases in introduction and conclusion. “Besides providing support for the short-term (dynamic) planning, the DSS can assist policy makers in getting a better understanding the possible consequences of (restricting) the availability of key-resources (e.g., vaccines, vaccination capacity and number of medical hubs)”

Author Response

On behalf of my coauthors, I would like to thank you for the opportunity to revise and resubmit our manuscript vaccines-1382518, entitled “A data-driven digital application to enhance the capacity planning of the COVID-19 vaccination process”.  We found your comments to be very helpful in revising the manuscript and have carefully considered and responded to each suggestion. In the majority of cases we were successful in incorporating the feedback into our revised manuscript.

See attachment for an overview of our response to the comments and the corresponding changes we made in the text.

Thank you again for your consideration of our revised manuscript.

Comment

Line Number

Response & corresponding changes

The text needs a language revision

We have taken the suggested "minor changes" to heart and implemented them (see below). Furthermore, a thorough spelling and grammar check has been carried out.

The graph in Figure 4 shows continuous behaviour over time and is apparently unnecessary.

We changed Figure 4 into lightmode and used non-horizontal lines (changed allocation to equal inside dashboard)

The article comes to a sudden end. The results presented in topic four can be improved, there is a lack of discussion, and the conclusion presents the same sentences as the introduction. I can think of some discussion questions. What are the advantages and disadvantages of the model? How do the authors imagine the application of the model by other countries? How might the model be appropriate for other diseases?

Same phrases in introduction and conclusion. “Besides providing support for the short-term (dynamic) planning, the DSS can assist policy makers in getting a better understanding the possible consequences of (restricting) the availability of key-resources (e.g., vaccines, vaccination capacity and number of medical hubs)”

The “Conclusion section” has been revised according to your comments:

Conclusions & discussion

We presented an advanced DSS that supports decision making regarding the capacity planning of the COVID-19 vaccination process. The DDS aims to minimize the per-class waiting time with respect to the vaccination locations and the distribution of the vaccines and healthcare professionals across the large-scale vaccination locations. Furthermore, the DSS-user is free to experiment with different starting positions and strategies, and as such, the DSS is ideally suited for providing support in a dynamic environment.

Besides providing support for the short-term (dynamic) planning, the DSS can assist policymakers in getting a better understanding the possible consequences of (restricting) the availability of key-resources (e.g., vaccines, vaccination capacity and number of medical hubs). The successful implementation in practice of the model presented in 4.1 shows that mathematical models can be useful in streamlining the last-mile of the COVID-19 vaccination process.

We emphasize that the approaches and DSS presented in this paper are also applicable way beyond the Dutch context. Hence, they can be applied to other countries, regions and/or large-scale vaccination campaigns worldwide. This is because the model is highly generic and the main characteristics of the last-mile (logistics) infrastructure are often similar. Nevertheless, the current version of the DSS is particularly useful in a dynamic context where circumstances are continuously changing (e.g., availability of vaccines of different types, changes in supply, the availability of healthcare professionals).

Despite being a valuable tool, some important limitations should be acknowledged. Transportation and storage constraints are not considered. In addition, the applied approach assumes a smooth vaccination process. Hence, stochasticity regarding e.g. last-mile delivery times and the vaccination procedure itself are not explicitly taken into account.

Finally, we address a number of topics for follow-up research. First, the inclusion of stochasticity which requires the use of historical data to estimate, e.g., the availability and amount of specific vaccine types and vaccination willingness. Second, the scalability of the algorithms can be improved by the development of sophisticated genetic algorithms, meta-heuristics or machine learning techniques; the advantage of those techniques is that the computation times remain acceptable even for large-scale model instances.

The authors could discuss the existing literature on large-scale vaccination programs (also considering other similar studies that are for other diseases). Suggested articles for discussion:

Rocha, T. A. H., Boitrago, G. M., Mônica, R. B., Almeida, D. G. D., Silva, N. C. D., Silva, D. M., ... & Vissoci, J. R. N. (2021). National COVID-19 vaccination plan: using artificial spatial intelligence to overcome challenges in Brazil. Ciencia & saude coletiva, 26, 1885-1898.

Krzysztofowicz, S., & OsiÅ„ska-Skotak, K. (2021). The Use of GIS Technology to Optimize COVID-19 Vaccine Distribution: A Case Study of the City of Warsaw, Poland. International Journal of Environmental Research and Public Health, 18(11), 5636.

Sun, X., Andoh, E. A., & Yu, H. (2021). A Simulation-Based Analysis for Effective Distribution of COVID-19 Vaccines: A Case Study in Norway. Transportation Research Interdisciplinary Perspectives, 100453.

Lemaitre, J. C., Pasetto, D., Zanon, M., Bertuzzo, E., Mari, L., Miccoli, S., ... & Rinaldo, A. (2021). Optimizing the spatio-temporal allocation of COVID-19 vaccines: Italy as a case study. medRxiv.

We included three of the suggested articles in the literature part of the introduction section:

In [9] the distribution of vaccines to vaccination sites is optimized using Geographic Information Systems (GIS) technology. By applying the developed methodology in the city of Warsaw in Poland, the authors demonstrated ways to achieve uniform vaccination coverage throughout the city. In the recent study of [10] a simulation-based approach is presented which combines route optimization and dynamic simulation to improve the logistics performance of the COVID-19 vaccine distribution. The result of their scenario analysis shows that service level, cost-effectiveness, environmental performance, and equity of a cold chain vaccine logistics system can be significantly influenced by the fleet size, the fleet composition, the type of vehicle used, and the route optimization. [11] developed a novel control framework to derive optimal vaccination strategies, whereby considering epidemiological projections and constraints on vaccine supply and distribution logistics.

Minor corrections

16

“And although some” to “Although some”

Minor corrections

21

“But how“ to “However, how”

Minor corrections

28

“required to make a distinction” to “required to distinguish”

Minor corrections

31

“own homes, in hospitals or the nursing homes” to “own homes, hospitals, or nursing homes”

Minor corrections

23

“In this study the focus lies” to “In this study, the focus lies”

Minor corrections

40-42

“Finally, given the large social impact that the COVID-19 vaccination program has, it is important that that the underlying processes and progress are as transparent as possible (see e.g., [3,5]).” to  “Finally, given the significant social impact of the COVID-19 vaccination program, the processes and underlying progress should be as transparent as possible (see, for example, [3,5]).”

Minor corrections

46

“In [6] a hybrid” we suggest inserting a comma “In [6], a hybrid”

Minor corrections

48

“Tavana et al. [7]” to [7]

Minor corrections

49

“enhance a more equitable distribution” we suggest “enhance more equitable distribution”

Minor corrections

56

“And although, in the past numerous” we suggest inserting a comma “Although, in the past, numerous”

Minor corrections

62

“which focus on the streamlining of the last mile of the COVID-19 vaccination programs are scarce.” we suggest “that focus on streamlining the last mile of the COVID-19 vaccination programs are scarce.”

Minor corrections

70

“policy makers” we suggest “policymakers” (check the entire text)

Minor corrections

70-71

“understanding into the possible” we suggest “understanding of the possible”

Minor corrections

73

“And although in this paper the Dutch situation is taken as point of departure” we suggest “Although the Dutch situation is taken as a point of departure in this paper,“

Minor corrections

76

“list which assigns different” we suggest “list that assigns different”

Minor corrections

86

“in the urban areas” we suggest “in urban areas”

Minor corrections

109

“With regard to the COVID-19 vaccination process it is” we suggest “Concerning the COVID-19 vaccination process, it is”

Minor corrections

119

“well known” we suggest “well-known”

Minor corrections

121

“Programming (ILP) the” we suggest inserting a comma “Programming (ILP), the”

Minor corrections

122

“the computation time of these approaches are” → “the computation time of these approaches is”

Minor corrections

122-123

“the large number” we suggest “a large number”

Minor corrections

140-141

“However as the algorithm” we suggest inserting a comma “However, as the “

Minor corrections

145

“we can conclude the algorithm” we suggest “we can conclude that the algorithm”

Minor corrections

152

“be finished with vaccinating earlier” we suggest “be finished vaccinating earlier”

Minor corrections

171

“professionals a short” we suggest inserting a comma “professionals, a short”

Minor corrections

187

“In case of a” we suggest “In case of the”

Minor corrections

209

“by two, in order to ensure” → “by two to ensure”

Minor corrections

154-155

“and are not considered in the division of vaccines anymore.” we suggest “and are no longer considered in the division of vaccines.”

Minor corrections

216

“In this section we present a DSS which can” we suggest “In this section, we present a DSS that can”

Minor corrections

219

“of the dashboard particular attention” we suggest inserting a comma “of the dashboard, particular attention”

Minor corrections

245-246

“However, to account for the continuously changing order of vaccination prioritization, this input parameter has been implemented.”

→ “However, this input parameter has been implemented to account for the continuously changing order of vaccination prioritization.”

Minor corrections

251

“as default value” we suggest “as the default value”

Minor corrections

295

​​“strategy effects” we suggest “strategy affect”

Minor corrections

320

“to assign” we suggest “to assigning”

Minor corrections

327

“time with respect to the” → “time to the”

Reviewer 2 Report

The paper is short and accomplishes what it proposes to research, however, I believe it could be improved by mentioning metaheuristics and other state-of-the-art algorithms for Combinatorial Optimization.

The used methods are not the top ones.

However, I believe it is an important topic and relevant to be published.

I suggest the author to improve the papers with workflows and figures for simplifing algorithms comprehension and also improve background on Optimization.

Author Response

On behalf of my coauthors, I would like to thank you for the opportunity to revise and resubmit our manuscript vaccines-1382518, entitled “A data-driven digital application to enhance the capacity planning of the COVID-19 vaccination process”.  We found your comments to be very helpful in revising the manuscript and have carefully considered and responded to each suggestion. Below you find an overview of our response to the comments and the corresponding changes we made in the text.

Thank you again for your consideration of our revised manuscript.

Comment 1:

The paper is short and accomplishes what it proposes to research, however, I believe it could be improved by mentioning metaheuristics and other state-of-the-art algorithms for Combinatorial Optimization.

Response & Changes

We have expanded the conclusion section with the following text:

Finally, we address a number of topics for follow-up research. First, the inclusion of stochasticity which requires the use of historical data to estimate, e.g., the availability and amount of specific vaccine types and vaccination willingness. Second, the scalability of the algorithms can be improved by the development of sophisticated genetic algorithms, meta-heuristics or machine learning techniques; the advantage of those techniques is that the computation times remain acceptable even for large-scale model instances. 

Comment 2:

I suggest the author to improve the papers with workflows and figures for simplifing algorithms comprehension and also improve background on Optimization.

Response & Changes

We carefully considered these comments and decided not to implement them. As the developed algorithms are not that complex, we believe the corresponding pseudo-codes are (relatively) easy to follow. Furthermore, the pseudo-codes provide the reader with details which are necessary to implement the algorithms in practice.

Reviewer 3 Report

Report on the paper "A data-driven digital application to enhance the capacity planning of the COVID-19 vaccination process"

Summary: This research presents an advanced decision support system (DSS) that aids decision-making in the capacity planning of the COVID-19 vaccination process. The DSS intends to reduce per-class waiting times in relation to the locations of medical hubs (i.e., vaccination locations) and the allocation of available vaccines and healthcare experts between the medical hubs, commencing with the national "vaccination priority list" (over time). The DSS is great for giving guidance in a constantly changing environment since it allows the user to experiment with different starting locations and techniques. Aside from assisting policymakers with short-term (dynamic) planning, the DSS can also help policymakers gain a better grasp of the potential effects of (restricting) key-resource availability (e.g., vaccines, vaccination capacity, and the number of medical hubs). Despite its usefulness, there are several significant drawbacks to be aware of. The adopted technique assumes a smooth immunization procedure and ignores transportation and storage restrictions. The model given in Equation (4.1) was successfully implemented in practice, demonstrating that mathematical models can be effective in expediting the last mile of the COVID-19 immunization process.

Evaluation: This paper is very clear, well written and interesting. The topic is of very high importance for the coming months/years. I am totally convinced by the considered modeling approach, which is conducted in a very professional way. Furthermore, its limitations are fairly discussed, and can inspire for possible improvements in future works. I have not detected things to correct. I recommend this work for publication, without reserve. 

Author Response

On behalf of my coauthors, I would like to thank you for reviewing our manuscript vaccines-1382518, entitled “A data-driven digital application to enhance the capacity planning of the COVID-19 vaccination process”.     Furthermore we would like to thank you for your positive assessment.

Round 2

Reviewer 1 Report

The authors did an excellent job.

Reviewer 2 Report

In Algorithm 1.

Line 3 sort the hubs.

It is not clear how line 4 pics the smallest since the vector is already sorted.

I do not agree that the proposed DSS is advanced, as mentioned on line 333, it is a set of constructive algorithms with direct rules.

I suggested, previously, that the authors should focus on metaheuristics background and expected that would evolve in that direction, a simple generation of distinct solutions would improve the algorithm used by generating multiple solutions considering an Alpha Parameters on line 3 of Algorithm 1. That parameter could guide how sorted will be this first sort.

Vide GRASP metaheuristic.

I suggest the author to make clear that the approach is an heuristic that is simple or to improve the algorithm in order to evolve towards the real difficulty of the problem.

Author Response

Dear reviewer,

Thank you for your very careful (second) review of our paper, and for the useful comments and suggestions. We believe that manuscript has been improved and hope it will be accepted for publication in Vaccines. Below we address all comments step by step. We uploaded a revised version of the manuscript in which the changes have been highlighted.

Kind regards,

The authors.

Comment 1:

In Algorithm 1.

Line 3 sort the hubs.

It is not clear how line 4 pics the smallest since the vector is already sorted.

Response:

Point 1 and 2 about algorithm 1 should be combined. Hence it is not necessary to state line 3 before line 4.  After all, in line 4 we select the smallest hub, so sorting (which is done in line 3) is not necessary. We agree and therefore removed line 3 from the pseudocode of algorithm 1.

Comment 2

I do not agree that the proposed DSS is advanced, as mentioned on line 333, it is a set of constructive algorithms with direct rules.

Response:

We agree and removed the word “advanced”.

Comment 3

I suggested, previously, that the authors should focus on metaheuristics background and expected that would evolve in that direction, a simple generation of distinct solutions would improve the algorithm used by generating multiple solutions considering an Alpha Parameters on line 3 of Algorithm 1. That parameter could guide how sorted will be this first sort.

Vide GRASP metaheuristic.

I suggest the author to make clear that the approach is an heuristic that is simple or to improve the algorithm in order to evolve towards the real difficulty of the problem.

Response:

We agree and included the following text in the “Conclusion & discussion” section:

Finally, we address a number of topics for follow-up research. Since the DSS in its current state uses relatively simple but fast heuristics, there is ample room for refinement of the model by applying meta-heuristics. Meta-heuristics are general-purpose optimization algorithms [22] where the term meta refers to the higher-level general methodologies, which are used to guide the underlying heuristic strategy [23]. Metaheuristics are highly diverse in nature, see [24][25][26] and references therein for overviews. In the context of the present paper, a highly promising type of meta-heuristics is offered by the concept of greedy randomized adaptive search procedures (GRASP), which is based on an iterative approach consisting of two phases: (1) a construction phase, and (2) a local search phase. We refer to [27] for details on GRASP.

Round 3

Reviewer 2 Report

Due to the error in Algorithm 1 regarding the sorting that is not really used I have not more confidence in judging this paper.

I suggest another reviewer and keep my major review as suggestion.